# Logits of API-Protected LLMs Leak Proprietary Information

**Matthew Finlayson   Xiang Ren   Swabha Swayamdipta**
Thomas Lord Department of Computer Science
University of Southern California
{mfinlays, xiangren, swabhas}@usc.edu

## Abstract

Large language model (LLM) providers often hide the architectural details and parameters of their proprietary models by restricting public access to a limited API. In this work we show that, with only a conservative assumption about the model architecture, it is possible to learn a surprisingly large amount of non-public information about an API-protected LLM from a relatively small number of API queries (e.g., costing under $1000 USD for OpenAI's gpt-3.5-turbo). Our findings are centered on one key observation: most modern LLMs suffer from a softmax bottleneck, which restricts the model outputs to a linear subspace of the full output space. We exploit this fact to unlock several capabilities, including (but not limited to) obtaining cheap full-vocabulary outputs, auditing for specific types of model updates, identifying the source LLM given a single full LLM output, and even efficiently discovering the LLM's hidden size. Our empirical investigations show the effectiveness of our methods, which allow us to estimate the embedding size of OpenAI's gpt-3.5-turbo to be about 4096. Lastly, we discuss ways that LLM providers can guard against these attacks, as well as how these capabilities can be viewed as a feature (rather than a bug) by allowing for greater transparency and accountability.

## 1   Introduction

As large language models (LLMs) become more capable and valuable, many LLM providers have shifted toward training closed-source production LLMs, accessible only via (paywall-restricted) APIs to protect trade secrets (e.g., OpenAI et al., 2024). As a useful feature, these APIs often allow users to access the probabilities (or logits, i.e., unnormalized scores) that the LLM assigns to specific tokens. It turns out that such an interface reveals much more information about the underlying model than previously thought, including exposing non-public architectural details of the LLM. Closed APIs, therefore, may provide a false sense of security to production LLM providers who may assume that their product information is private. At the same time, these capabilities empower the community with tools to audit LLM providers for bad behaviors such as unannounced model updates and abuse of open-source LLMs (Mökander et al., 2023).

In this paper we show that is possible to extract detailed information about LLM parameterization using only common API configurations. Our findings are centered on one key observation: most modern LLMs suffer from a softmax bottleneck (Yang et al., 2018), which restricts the model outputs to a linear subspace of the full output space, as illustrated in Figure 1. We call this restricted output space the LLM's *image* (§2). This image can be obtained by collecting a small number of LLM outputs, and serves as a unique identifier or a *signature* for the model. We propose several algorithms that enable us to obtain the LLM image at low cost and speed for standard LLM APIs (§3).

Furthermore, we show that LLM images are useful in several applications, revealing important information about the model architecture. Concretely, we can exploit this image to provide an efficient algorithm for extracting full LLM outputs (§4), to find the hidden dimension (embedding) size of the LLM (§5), to detect and identify different kind of updates made to the model and to attribute model outputs to a specific model (§6), among other

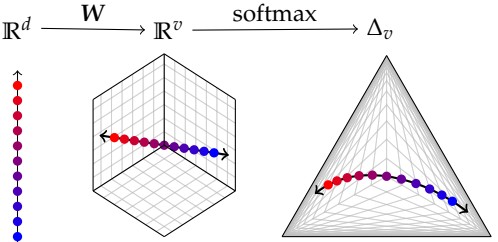

Figure 1: LLM outputs are constrained to a low-dimensional subspace of the full output space. We can use this fact to glean information about API-protected LLMs by analyzing their outputs. Here we show how a toy LLM's low-dimensional embeddings in $\mathbb{R}^d$ (illustrated here as a 1-D space) are transformed linearly into logits in $\mathbb{R}^v$ (here, a 3D space) via the softmax matrix $W$. The resulting outputs lie within a ($d = 1$)-dimensional subspace of the output space. We call this low-dimensional subspace the *image* of the model. We can obtain a basis for the image of an API-protected LLM by collecting $d$ of its outputs. The LLM's image can reveal non-public information, such as the LLM's embedding size, but it can also be used for accountability, such as verifying which LLM an API is serving.

potential applications (§7). We demonstrate several of our proposed applications in order to empirically verify their effectiveness. Notably, we design and implement an algorithm for finding the embedding size of an API-based LLM, and use it to estimate the embedding size of gpt-3.5-turbo (a closed-source API-protected LLM) to be 4096.

Overall, our proposed methods have benefits for both LLM providers and for their clients. LLM providers may use their model images to establish unique identities for their models, thereby protecting their product and building trust with clients. Our proposed methods will also allow LLM clients to hold providers accountable for malicious behavior (Anderljung et al., 2023). As a concrete use case in accountability, we demonstrate, using checkpoints from open-source LLMs that our LLM images can be used to attribute LLM output probabilities (or logits) to their generating model with high accuracy. The sensitivity of our LLM image to slight changes in the LLM parameters also makes them suitable for inferring granular information about the specific type of model update.

Considering several proposals to guard access to LLM images, we find no obvious fix without dramatically altering the LLM architecture or making the API considerably less useful (§8). Providers who choose to alter their API to prevent LLM image access risk removing interfaces with valuable and safe use cases for LLM clients. Though our findings could be viewed as a bug that LLM providers might feel compelled to patch, we prefer to view them as *features* that LLM providers may choose to keep in order to better maintain trust with their customers by allowing outside observers to audit their model. Ultimately, our results serve as a recommendation to LLM providers to carefully consider the consequences of their LLM architectures and APIs.

This paper's contributions include

- A method for extracting information about API-protected models, including the model's output space and embedding size.
- Methods for extracting full-vocabulary logprob outputs from LLM APIs.
- An estimate of the embedding size of an API-protected LLM (gpt-3.5-turbo).
- An accelerated logprob extraction algorithm based on the LLM image.
- An exploration of several other applications of our method for model accountability.

Concurrently with our work, Carlini et al. (2024) propose a very similar approach for exposing details of production LLMs, though with a focus on defenses and mitigations against such attacks. The "final layer" that they extract in their attack corresponds to what we refer to in our paper as the *model image*. We view our papers as complementary, since our work emphasises practical applications of LLM images for better LLM accountability.

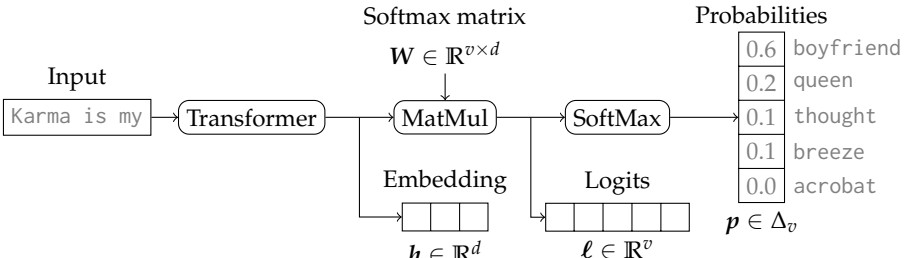

Figure 2: A typical language model architecture. After the input its processed by a neural network, usually a transformer (Vaswani et al., 2017), into a low-dimensional embedding $h$, it is multiplied by the softmax matrix $W$, projecting it linearly from $\mathbb{R}^d$ onto $\mathbb{R}^v$ to obtain the logit vector $\ell$. The softmax function is then applied to the logit vector to obtain a valid probability distribution $p$ over next-token candidates.

## 2 LLM outputs are restricted to a low-dimensional linear space

The outputs of typical LLMs are naturally constrained to lie on a $d$-dimensional subspace of the full output space (Yang et al., 2018; Finlayson et al., 2024). To understand this, begin by considering the architecture of a typical LLM (Figure 2). In this architecture, a transformer[1] with embedding size $d$ outputs a low-dimensional *contextualized embedding* $h \in \mathbb{R}^d$ (or simply *embedding*). Projecting the embedding onto $\mathbb{R}^v$ via the linear map defined by the LLM's *softmax matrix* $W$, we obtain *logits* $\ell = Wh$.

**Theorem 1** (Low-rank logits). *LLM logits lie on a $d$-dimensional subspace of $\mathbb{R}^v$.*

*Proof.* Because $W$ is in $\mathbb{R}^{v \times d}$, its rank (i.e., number of linearly independent columns) is at most $d$. The rank of a matrix corresponds to the dimension of the *image* of the linear map it defines, i.e., the vector space comprising the set of possible outputs of the function. In other words, if the linear map $w$ is defined as $w(h) = Wh$, then $w$'s image $\text{im}(w) = \{w(h) \in \mathbb{R}^v : h \in \mathbb{R}^d\}$ is a $d$-dimensional subspace of $\mathbb{R}^v$. □

Thus, the LLM's logits will always lie on the $d$-dimensional[2] subspace of $\mathbb{R}^v$.

The low-dimensional restriction of logits actually translates into a similar restriction on LLM probabilities, which is useful since most LLM APIs return (log-)probabilities instead of logits. To understand why this is, we turn our attention to the model's next-token distribution $p = \text{softmax}(\ell)$. The softmax function[3] transforms logits into a valid probability distribution, i.e., a $v$-tuple of real numbers between 0 and 1 whose sum is 1. The set of valid probability distributions over $v$ items is commonly referred to as the *$v$-simplex*, or $\Delta_v$.

**Theorem 2** (Low-rank probabilities). *LLM probabilities lie on a $d$-dimensional subspace of $\Delta_v$.*

*Proof.* Perhaps surprisingly, $\Delta_v$ is also a valid $(v-1)$-dimensional vector space (albeit under non-standard definitions of addition and scalar multiplication) and the softmax function is a linear map $\mathbb{R}^v \to \Delta_v$ (Aitchison, 1982; Leinster, 2016). Therefore, since $\dim(\text{im}(w)) = d$, we also know that $\text{im}(\text{softmax} \circ w)$ is a $d$-dimensional subspace of $\Delta_v$. □

Thus, LLM output probabilities must also reside in a $d$-dimensional subspace.

Finally, since $\Delta_v$ is an unconventional vector space, it is useful to translate LLM probability distributions into a more conventional vector space. To do so, we use the fact that

---

[1]Technically, any neural network suffices here.

[2]More accurately, the logits will always lie on an *at-most-$d$*-dimensional subspace. For convenience, we assume full-rank matrices, and thus a $d$-dimensional subspace.

[3]Our use of "softmax" here corresponds to *temperature softmax* with temperature 1.

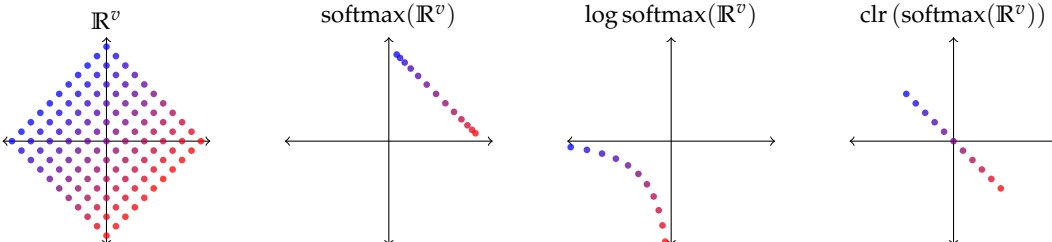

Figure 3: Points in the logit space $\mathbb{R}^v$ (far left) are mapped via the softmax function to points (probability distributions) on the simplex $\Delta_v$ (middle left). Crucially, the softmax maps all points that lie on the same diagonal (shown as points of the same color) to the same probability distribution. For numerical stability, these values are often stored as log-probabilities (middle right). The clr transform returns probability distributions to points to a subspace $U_v$ of the logit space (far right). The softmax function and clr transform are inverses of one another, and form an isomorphism between $U_v$ and $\Delta_v$.

$\Delta_v$ is isomorphic to a vector subspace of $\mathbb{R}^v$ via the softmax function. In particular, Figure 3 illustrates how $\Delta_v$ is isomorphic to the hyperplane $U_v$ that is perpendicular to the all-ones vector $\mathbf{1}_v$. The isomorphism's inverse mapping $\Delta_v \to U_v$ is the *center log ratio transform* $\mathrm{clr}(\boldsymbol{p}) = \log \boldsymbol{p} - \frac{1}{v} \sum_{i=1}^{v} \log p_i$. By linearity, $\mathrm{im}(\mathrm{clr} \circ \mathrm{softmax} \circ w)$ is a $d$-dimensional subspace of $U_v \subset \mathbb{R}^v$.

Thus, LLM outputs occupy $d$-dimensional subspaces of logit space $\mathbb{R}^v$, probability space $\Delta_v$, and $U_v$. We call these subspaces the *image* of the LLM on each given space. A natural consequence this low-dimensionality is that any collection of $d$ linearly independent LLM outputs form a basis for the image of the model, i.e., all LLM outputs can be expressed as a linear combination of these outputs.

## 3   Obtaining full outputs from API-protected LLMs

As explained in the previous section, in order to obtain the image of an LLM, we need to collect outputs (token probabilites for each token in the vocabulary) from the LLM. Unfortunately, most LLM APIs do not return full outputs, likely because full outputs are large and expensive to send over an API, but perhaps also to prevent API abuse, since full LLM outputs contain lots of useful information (Dosovitskiy and Brox, 2016; Morris et al., 2023) and can be used to distill models (e.g., Hinton et al., 2015; Hsieh et al., 2023). In their paper, Morris et al. (2023) give an algorithm for recovering full outputs from restricted APIs by taking advantage of a common API option that allows users to add a bias term $\beta \leq \beta_{\max}$ to the logits for specific tokens. The algorithm they describe requires $v \log(\beta/\epsilon)$ calls to the API to obtain one full output with precision $\epsilon$, or exactly $v$ calls when the API returns log-probabilities of the top-2 tokens. Carlini et al. (2024) also propose a logprob-free method based on the same principle.

We give a variant of their algorithm for APIs that return the log-probability of the top-$k$ tokens that obtains full outputs in $v/k$ API calls. We find that this improved algorithm suffers from numerical instability, and give a numerically stable algorithm that obtains full outputs in $v/(k-1)$ API calls. We also give a practical algorithm for dealing with stochastic APIs that randomly choose outputs from a set of $n$ possible outputs, a behavior we observe in OpenAI's API. This algorithm allows the collection of full outputs in $nv/(k-2)$ API calls on average. Table 1 gives an overview of our algorithms with back-of-the envelope cost estimates for a specific LLM.

### 3.1   Full outputs from APIs with logprobs

Our goal is to recover a full-vocabulary next-token distribution $\boldsymbol{p} \in \Delta_v$ from an API-protected LLM. We will assume that the API accepts a prompt on which to condition the

| Algorithm | Complexity | API calls per output | Image price (USD) |
|---|---|---|---|
| Logprob-free (Morris et al., 2023) | $v \log(\beta_{\max}/\epsilon)$ | 800 000 | 16 384 |
| With logprobs | $v/k$ | 20 000 | 410 |
| Numerically stable | $v/(k-1)$ | 25 000 | 512 |
| Stochastically robust | $nv/(k-2)$ | 133 000 | 2724 |
| LLM Image (§4) | $O(d)$ | 800 | – |

Table 1: A summary of our proposed algorithms for obtaining full LLM outputs, with estimates for the number of API calls required per output, and the price of acquiring the model image. Estimates are based on a `gpt-3.5-turbo`-like API LLM with $v = 100\,000$, $d = 4096$, $\epsilon = 10^{-6}$, $k = 5$, $\beta_{\max} = 100$, and $n = 4$. Note that the $O(d)$ algorithm cannot be used to obtain the LLM image, since it relies on having LLM image as a preprocessing step.

distribution, as well as a list of up to $k$ tokens and a bias term $\beta \leq \beta_{\max}$ to add to the logits of the listed tokens before applying the softmax function. The API returns a record with the $k$ most likely tokens and their probabilities from the biased distribution. For instance, querying the API with $k$ maximally biased tokens, which (without loss of generality) we will identify as tokens $1, 2, \ldots, k$, yields the top-$k$ most probable tokens from the *biased* distribution $\boldsymbol{p}' = \text{softmax}(\boldsymbol{\ell}')$ where

$$\ell_i' = \begin{cases} \ell_i + \beta_{\max} & i \in \{1, 2, \ldots, k\} \\ \ell_i & \text{otherwise} \end{cases} \tag{1}$$

and $\boldsymbol{\ell} \in \mathbb{R}^v$ is the LLM's logit output for the given prompt.

Assuming that the logit difference between any two tokens is never greater than $\beta_{\max}$, these top-$k$ biased probabilities will be $p_1', p_2', \ldots, p_k'$. For each of these biased probabilities $p_i'$, we can solve for the unbiased probability as

$$p_i = \frac{p_i'}{\exp \beta_{\max} - \exp \beta_{\max} \sum_{j=1}^{k} p_j' + \sum_{j=1}^{k} p_j'} \tag{2}$$

(proof in the §A.1). Thus, for each API call, we can obtain the unbiased probability of $k$ tokens, and obtain the full distribution in $v/k$ API calls.

### 3.2 Numerically stable full outputs from APIs

In practice, the algorithm described in §3.1 suffers from severe numerical instability, which can be attributed to the fast-growing exponential term $\exp \beta_{\max}$, and the term $\sum_{j=1}^{k} p_j'$ which quickly approaches 1. We can eliminate the instability by sacrificing some speed and using a different strategy to solve for the unbiased probabilities. Without loss of generality, let $p_v$ be the maximum unbiased token probability. This can be obtained by querying the API once with no bias. If we then query the API and apply maximum bias only to tokens $1, 2, \ldots, 1 - k$, then the API will yield $p_1', p_2', \ldots, p_{k-1}'$ and $p_v'$. We can then solve for the unbiased probabilities of the $k - 1$ tokens

$$p_i = \exp(\log p_i' - \beta_{\max} - \log p_v' + \log p_v) \tag{3}$$

(proof in §A.2). By finding $k - 1$ unbiased token probabilities with every API call, we obtain the full output in $v/(k-1)$ calls total. Algorithm 1 gives a formal description of this method.

### 3.3 Full outputs from stochastic APIs

Each of the above algorithms assume that the API is deterministic, i.e., the same query will always return the same output. However, this may not always be the case; for instance, we find that OpenAI's LLM APIs are *stochastic*. While this would seem to doom any attempt at obtaining full outputs from the LLM, it is possible to counter certain types of stochasticity. In particular, we model stochastic API behavior as a collection of $n$ outputs $\boldsymbol{p}^1, \boldsymbol{p}^2, \ldots, \boldsymbol{p}^n$

---

**Algorithm 1** Our numerically stable probability extraction algorithm, which takes a set of contexts, a set of tokens, and an API for an LLM with vocabulary $\mathcal{V}$ and returns the LLM's probabilities for each token in each context. The API takes a context and a set of token-bias pairs and returns the top-$k$ biased probabilities. $\mathcal{P}(S)$ denotes the power set of a set $S$.

---

**function** EXTRACT( contexts $\subseteq \mathcal{V}^*$, tokens $\subseteq \mathcal{V}$, API : $\mathcal{V}^* \times \mathcal{P}(\mathcal{V} \times \mathbb{R}) \to \mathcal{P}(\mathcal{V} \times \mathbb{R})$ )

    probs $\in \mathbb{R}^{|\text{contexts}| \times |\text{tokens}|}$             ▷ Initialize empty index to collect probabilities

    **for** $c \in$ contexts **do**

        $(v, p_v) = \text{argmax}_{(i,p) \in \text{API}(c, \varnothing)}\, p$            ▷ Get top probability token

        **for** each batch $T \subseteq$ tokens with $|T| = k - 1$ **do**     ▷ Partition tokens into batches

            bias $= \{(i, \beta)\}_{i \in T}$               ▷ Set biases to $\beta$ for batch tokens

            $\{(v, p_v')\} \cup \{(i, p_i')\}_{i \in T} = \text{API}(c, \text{bias})$       ▷ Get biased probabilities

            **for** $i \in T$ **do**

                probs$_{c,i} = \exp(\log p_i' - \beta - \log p_v' + \log p_v)$     ▷ Get unbiased probability

            **end for**

        **end for**

    **end for**

    **return** probs

**end function**

---

from which the API randomly returns from. This might be the result of multiple instances of the LLM being run on different hardware which results in slightly different outputs. Whichever instance the API returns from determines which of the $n$ outputs we get. In order to determine which of the outputs the API returned from, let $p_{v-1}^i$ be the second highest token probability for output $\boldsymbol{p}^i$, and observe that $\log p_v^i - \log p_{v-1}^i = \log p_v^{i\prime} - \log p_{v-1}^{i\prime}$ for all outputs $\boldsymbol{p}^i$ and biased outputs $\boldsymbol{p}^{i\prime}$ where tokens $v$ and $v-1$ are not biased (proof in §A.3). Therefore, by biasing only $k-2$ tokens for each call, the API will return $p_v^{i\prime}$ and $p_{v-1}^{i\prime}$, which we can use to find $\log p_v^i - \log p_{v-1}^i$, which serves as a unique identifier for the distribution. Thus, after an average of $nv/(k-2)$ calls to the API we can collect the full set of probabilities for at least one of the outputs.

## 4   Fast, full outputs using the LLM image

The dominating factor in runtime for the algorithms in §3 is the vocabulary size $v$, which can be quite large (e.g., over $100\,000$ for OpenAI LLMs). We now introduce a preprocessing step that takes advantage of the low-dimensional LLM output space to obtain $O(d)$ versions of all the above algorithms. Since $d \ll v$ for many modern language models, this modification can result in multiple orders of magnitude speedups, depending on the LLM. For instance, the speedup for a model like `pythia-70m` (Biderman et al., 2023) would be $100\times$. The key to this algorithm is the observation from §2 that $d$ linearly independent outputs from the API constitute a basis for the whole output space (since the LLM's image has dimension $d$). We can therefore collect these outputs $\boldsymbol{P} = \begin{bmatrix} \boldsymbol{p}^1 & \boldsymbol{p}^2 & \cdots & \boldsymbol{p}^d \end{bmatrix} \in \Delta_v^d$ as a preprocessing step in $O(vd)$ API calls using any of the above algorithms and $d$ unique prompts, and then use these to reconstruct the full LLM output after only $O(d)$ queries for each subsequent output.

To get a new full output $\boldsymbol{p}$, use any of the above algorithms to obtain $p_1, p_2, \ldots, p_d$. Since $\boldsymbol{p}$ resides in a $d$-dimensional space spanned by the columns of $\boldsymbol{P}$, the rest of the values of $\boldsymbol{p}$ are fully determined by these first $d$ values. §B gives the details of how to solve for the remaining values. Thus we can retrieve $\boldsymbol{p}$ in only $O(d)$ API queries.

This $(v/d)\times$ speedup makes any method that relies on full model outputs significantly cheaper so long as the number of model outputs needed exceeds $d$. Such methods include model stealing (Tramèr et al., 2016; Krishna et al., 2019) which attempts to learn a model that exactly replicates the behavior of a target model, and LM inversion (Morris et al., 2023) which uses LLM outputs to reconstruct hidden prompts. Additionally, the preprocessing

**Algorithm 2** Our algorithm for finding the model image and embedding size for an API-protected LLM. To get the embedding size of the model without finding the image, extract only the first $t$ token probabilities at each step. With API caching this takes $O(vd/k)$ time, or $O(d^2/k)$ to only extract the embedding size.

> **function** GETIMAGE( API : $\mathcal{V}^* \times \mathcal{P}(\mathcal{V} \times \mathbb{R}) \to \mathcal{P}(\mathcal{V} \times \mathbb{R})$ )
>     **for** $t = 1, 2, \ldots$ **do**
>         $L_t = \text{clr}\left(\text{EXTRACT}([t], \mathcal{V}, \text{API})\right) \in \mathbb{R}^{t \times |\mathcal{V}|}$       ▷ Get full logits for $t$ contexts
>         **if** $\text{rank}(L_t) = t - 100$ **then**       ▷ When the output rank stops increasing
>             **return** $L_{t-100}$     ▷ Return the model image. Embedding size is $t - 100$.
>         **end if**
>     **end for**
> **end function**

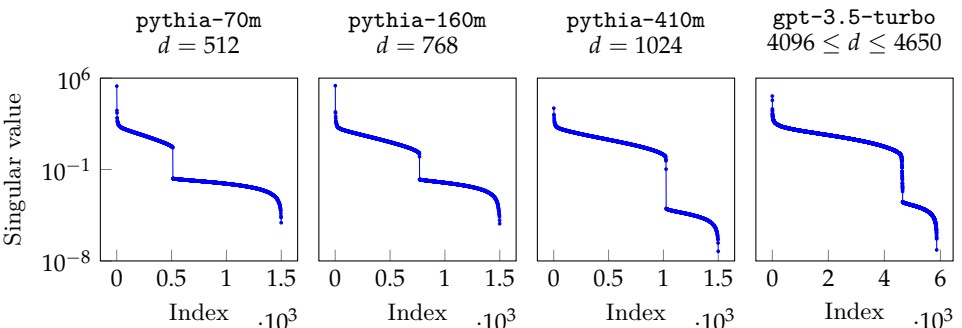

Figure 4: The singular values of outputs from LLMs with various known and unknown embedding sizes $d$. For each model with known embedding size, there is a clear drop in magnitude at singular value index $d$, indicating the embedding size of the model. Using this observation, we can guess the embedding size of `gpt-3.5-turbo` to be 4096.

step can be computed once then shared between any number of clients, further diluting the cost of full outputs.

## 5   Discovering the embedding size of API-protected LLMs

Assuming only the generic output layer described in Figure 2, it is possible to infer the embedding size $d$ of an API-protected LLM from its outputs alone. Since the model outputs occupy a $d$-dimensional subspace of $\Delta_v$, collecting $d$ linearly independent outputs $p^1, p^2, \ldots, p^d$ from the LLM forms a basis for the LLM's image. In other words, subsequent model outputs will be a linear combination of the first $d$ outputs. We can therefore discover the value of $d$ by collecting outputs one at a time until the number of linearly independent outputs in the collection (i.e., the rank) stops increasing, which will occur after we have collected $d$ prompts. We find that it suffices to slightly over-collect, e.g., $d + 1000$ outputs. Algorithm 2 formalizes this procedure.

To find the rank of model outputs, we use the fact that a matrix with $d$ linearly independent columns will have $d$ non-zero singular values. We use singular value decomposition to obtain the singular values of the matrix $L = \begin{bmatrix} \text{clr}(p^1) & \text{clr}(p^2) & \cdots & \text{clr}(p^{d+1000}) \end{bmatrix}$ and observe the index at which the magnitude of the singular values drops to zero, which occurs at index $d$. In practice, numerical imprecision causes the magnitudes drop *almost* to zero.

To validate our method, we collect next-token distributions (conditioned on unique, 1-token prompts) from several open-source LLMs from the Pythia family (Biderman et al., 2023) with embedding sizes ranging from 512 to 1024. For all these models, the singular values of the resulting output matrix drop precisely at index $d$, as shown in Figure 4.

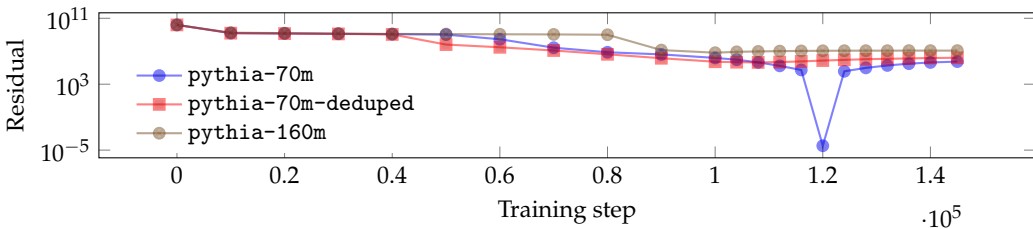

Figure 5: Residuals of the least-squares solution of $Lx = \ell$ for an output $\ell$ from the `pythia-70m` checkpoint at training step 120 000, and output matrices $L$ from various Pythia model checkpoints. High residuals indicate that the output is not in a model's image.

To demonstrate our method's effectiveness against API-protected LLMs, we use our algorithm from §3.3 to collect nearly 6,000 next-token distribution outputs from `gpt-3.5-turbo`[4], a popular API-protected LLM whose embedding size is not publicly disclosed. We find that this model's singular values drop between index 4,600 and 4,650, indicating that the embedding size of this model is at most this size. This predicted embedding size is somewhat abnormal, since LLM embedding sizes are traditionally set to powers of two. If this were the case for `gpt-3.5-turbo`, it would be reasonable to guess that the embedding size is $2^{12} = 4096$. We predict that our raw estimate of 4600–4650 is an overestimate of the true embedding size, since any abnormal outputs due to errors (whether in our own code or OpenAI's) would only increase the dimensionality of the observed output space. For instance, if we inadvertently collected 504 corrupted outputs, then a model with embedding size 4096 would appear to have an embedding size of 4600.

Knowing the embedding size alone is insufficient to ascertain an LLM's parameter count. However, since most known transformer-based LLMs with embedding size 4096 have around 7 billion parameters, it is likely that `gpt-3.5-turbo` has a similar number of active parameters. Any other parameter count would result in either abnormally narrow or wide models, which often perform worse. The actual parameter count may be much higher if the model uses a "mixture-of-experts" architecture (Shazeer et al., 2017). Given the active development and decreasing cost of inference with `gpt-3.5-turbo`, it is possible that its size and architecture has changed over time. Fortunately, our method can be used to monitor these updates over time, alerting end-users when LLM providers change embedding size (and presumably therefore, model size).

## 6 Attributing model outputs and auditing model updates

An LLM image can serve as an identifying signature for the model, as shown in Figure 5, where the logit output from a `pythia-70m` checkpoint lies uniquely in the checkpoint's image, and not in the image of the preceding or following checkpoints, nor the checkpoints of any other similar model.[5] This suggests that LLM images are highly sensitive to parameter changes, which makes sense because the intersection of two different $d$-dimensional spaces is an even lower-dimensional space. Thus, it is possible to determine precisely which LLM produced a particular full-vocabulary output using only API access to a set of LLMs, and notably, without knowing the exact inputs to the model.

This finding has implications for LLM auditing (Mökander et al., 2023), such as obtaining granular information about updates to API-protected LLMs. For instance, if the LLM image remains the same but the logit outputs change, this would indicate a partial model update where some part of the model changes but the softmax matrix remains the same. This might happen when the LLM has a hidden prefix added to all prompts and this hidden prefix changes, or when some part of the model was updated while leaving the softmax

---

[4]We specifically use model version 0125, accessed February 1–19, 2024.

[5]Open source models with detailed checkpoint information (e.g., Biderman et al., 2023; Liu et al., 2023) make this type of detailed analysis possible.

| Change | Interpretation |
|---|---|
| No logit change, no image change | No update |
| Logit change, no image change | Hidden prompt change or partial finetune |
| Low-rank image change (§7.1) | LoRA update |
| Image change | Full finetune |

Table 2: Implications of image/logit changes.

matrix unchanged. Table 2 gives an overview for how to interpret various combinations of detectable API changes.

Using the same principle, one can also detect malicious use of open-source LLMs. If a provider attempts to profit off of an open-source model with a non-commercial license, an auditor can reveal the scheme by checking the LLM image, even if the provider used a hidden prompt to mask the model identity. It would be unlikely for images of two different LLMs to match, unless purposely designed to do so. Note that this test is one-sided, and malicious providers may slightly fine-tune an LLM to avoid detection.

# 7 More Applications

Access to the LLM's image unlocks several additional capabilites, some of which we review below. We leave further investigation of these methods for future work.

## 7.1 Detecting LoRA updates

Access to the LLM image can afford even finer granularity insight into LLM updates. For instance, as LoRA (Hu et al., 2022) is a popular parameter-efficient fine-tuning method which adjusts model weights with a low-rank update $AB$ where $A \in \mathbb{R}^{v \times r}$ and $B \in \mathbb{R}^{r \times d}$ so that the softmax matrix $W \in \mathbb{R}^{v \times d}$ becomes $W + AB$. It may be possible to detect these types of updates by collecting LLM outputs before ($L \in \mathbb{R}^{v \times d}$) and after ($L' \in \mathbb{R}^{v \times d}$) the update and decomposing them as $WH = L$ and $(W + AB)H' = L'$ where $H, H' \in \mathbb{R}^{d \times d}$. If such a decomposition is found then it is likely that the weights received a low-rank update. We leave it to future work to find an efficient algorithm for this decomposition.

## 7.2 Finding unargmaxable tokens

Due to the low-rank constraints on LLM outputs, it is possible that some tokens become *unargmaxable* (Demeter et al., 2020; Grivas et al., 2022), i.e., the token always has less probability than some other token. This happens when the LLM's embedding representation of the token lies within the convex hull of the other tokens' embeddings. Previously, finding unargmaxable tokens appeared to require full access to the softmax matrix $W$, however, it is possible to identify unargmaxable tokens using only the LLM's image, which our method is able to recover. This technique allows API clients to find tokens that the model is unable to output and thereby elicit unexpected model behavior.

## 7.3 Recovering the softmax matrix from outputs

One might use the image to approximately reconstruct the output layer parameters. We hypothesize that LLM embeddings generally lie near the surface of a hypersphere in $\mathbb{R}^d$ with a small radius $r$. We see evidence of this in the fact that the Pythia LLM embedding norms are all small and roughly normally distributed, as shown in Figure 6 in the Appendix. We can attempt to recover $W$ up to a rotation by assuming that all embeddings must have unit magnitude, then, given a matrix $L \in \mathbb{R}^{n \times v}$ of model logits, we can find $W$ (up to a rotation) by finding a decomposition $WH = L$ such that for all $i$, $\|W_i\|_2 = 1$. This solution may also be approximated by finding the singular value decomposition $W\Sigma V^\top$ of $L$, though rows of this $W$ will have magnitude less than 1.

### 7.4 Basis-aware sampling

In a recent paper, Finlayson et al. (2024) propose a decoding algorithm that avoids type-I sampling errors by identifying tokens that must have non-zero true probability. Importantly, this method relies on knowing the basis of the LLM's output space, and is therefore only available for LLMs whose image is known. Our approach for finding the model image makes this decoding algorithm possible for API LLMs.

## 8 Mitigations

We consider three proposals that LLM providers may take to guard against our methods. The first proposal is to limit or entirely remove API access to logprobs. This is theoretically insufficient, as Morris et al. (2023) show that it is possible to obtain full outputs using only information about the biased argmax token, albeit inefficiently in $v \log(\beta/\epsilon)$ API calls. Providers may rely on the extreme inefficiency of the algorithm to protect the LLM, as OpenAI appeared to do in response to Carlini et al. (2024). However, our algorithm in §4 brings the cost down to a feasible $O(d \log(\beta/\epsilon))$ API calls per output once the initial work of finding the LLM image finishes, the result of which can be made public and reused.

The second proposal is to remove API access to logit bias. This would be effective, since there are no known methods to recover full outputs from such an API. However, logit bias has several useful applications, such as for blocking undesirable tokens and controlling text generation, and making such a restriction could seriously degrade the usefulness of the API.

Lastly, we consider alternative LLM architectures that do not suffer from a softmax bottleneck. There are several such proposed architectures with good performance (e.g., Xue et al., 2021; Yu et al., 2023; Wang et al., 2024). Though this is the most expensive of the proposed defenses, due to the requirement of training a new LLM, it would have the beneficial side effect of also treating other tokenization issues that plague large-vocabulary LLMs (e.g., Itzhak and Levy, 2022). A transition to softmax-bottleneck-free LLMs would fully prevent our attack, since the model's image would be the full output space.

## 9 Conclusion

We have shown how the low-rank constraints imposed by the softmax bottleneck expose non-public information about API-protected LLMs. We also demonstrated how this information can be used to efficiently extract full model outputs, expose model hyperparameters, function as a unique model signature for auditing purposes, and even detect specific types of model updates, among other uses. We find that some current protections against these attacks are insufficient, and that the more effective guards tend to inhibit sanctioned API use cases (i.e., logit bias), or are expensive to implement (i.e., changing model architecture).

Overall, we find that the benefits of our proposed methods outweigh the harms to LLM providers. For instance, allowing LLM API users to detect model changes builds trust between LLM providers and their customers, and leads to greater accountability and transparency for the providers. Our method can also be used to implement efficient protocols for model auditing without exposing the model weights. On the other hand, discovering the embedding size of the LLM does not enable a competitor to fully recover the parameters of the LLM's softmax matrix or boost performance of their own model. Even efficiently extracting full outputs for model stealing or inversion (§4) is unlikely to have detrimental effects, as these are already known threats and are only made more efficient by our methods. We therefore believe that our proposed methods and findings do not necessitate a change in LLM API best practices, but rather expand the tools available to API customers, while informing LLM providers of what information their APIs expose.

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

## A Proofs

This appendix contains derivations for the equations used to solve for unbiased token probabilities given biased LLM outputs.

### A.1 Fast logprobs algorithm

Our goal is to prove

$$p_i = \frac{p_i'}{\exp \beta_{\max} - \exp \beta_{\max} \sum_{j=1}^{k} p_j' + \sum_{j=1}^{k} p_j'} \tag{4}$$

*Proof.* We begin with the definition of the softmax function

$$p_i' = \frac{\exp(\ell_i + \beta_{\max})}{\sum_{j=1}^{k} \exp(\ell_j + \beta_{\max}) + \sum_{j=k+1}^{v} \exp \ell_j} \tag{5}$$

We then rearrange to obtain

$$\sum_{j=k+1}^{v} \exp \ell_j = \frac{\exp(\ell_i + \beta_{\max})}{p_i'} - \sum_{j=1}^{k} \exp(\ell_j + \beta_{\max}), \tag{6}$$

the left hand side of which is independent of the bias term, meaning it is equivalent to when $b_{\max} = 0$, i.e.,

$$\frac{\exp \ell_i}{p_i} - \sum_{j=1}^{k} \exp \ell_j = \frac{\exp(\ell_i + \beta_{\max})}{p_i'} - \sum_{j=1}^{k} \exp(\ell_j + \beta_{\max}). \tag{7}$$

We can now rearrange

$$\frac{\exp \ell_i}{p_i} = \frac{\exp(\ell_i + \beta_{\max})}{p_i'} - \sum_{j=1}^{k} \exp(\ell_j + \beta_{\max}) + \sum_{j=1}^{k} \exp \ell_j \tag{8}$$

$$p_i = \frac{\exp \ell_i}{\frac{\exp(\ell_i + \beta_{\max})}{p_i'} - \sum_{j=1}^{k} \exp(\ell_j + \beta_{\max}) + \sum_{j=1}^{k} \exp \ell_j} \tag{9}$$

and expand

$$p_i = \frac{p_i' \exp \ell_i}{\exp(\ell_i + \beta_{\max}) - p_i' \sum_{j=1}^{k} \exp(\ell_j + \beta_{\max}) + p_i' \sum_{j=1}^{k} \exp \ell_j} \tag{10}$$

$$= \frac{p_i'^2 \exp(-\beta_{\max}) \exp(\ell_i + \beta_{\max})}{\exp(\ell_i + \beta_{\max}) - p_i' \sum_{j=1}^{k} \exp(\ell_j + \beta_{\max}) + p_i' \exp(-\beta_{\max}) \sum_{j=1}^{k} \exp(\ell_j + \beta_{\max})} \tag{11}$$

and finally simplify by multiplying the top and bottom of the right hand side by

$$\frac{1}{\sum_{j=1}^{k} \exp(\ell_j + \beta_{\max}) + \sum_{j=k+1}^{v} \exp \ell_j} \tag{12}$$

and which converts each term of the form $\exp(\ell_i + b_{\max})$ to $p_i'$, resulting in

$$p_i = \frac{p_i'^2 \exp(-\beta_{\max})}{p_i' - p_i' \sum_{j=1}^{k} p_j' + p_i' \exp(-\beta_{\max}) \sum_{j=1}^{k} p_j'} \tag{13}$$

$$= \frac{p_i' \exp(-\beta_{\max})}{1 - \sum_{j=1}^{k} p_j' + \exp(-\beta_{\max}) \sum_{j=1}^{k} p_j'} \tag{14}$$

$$= \frac{p_i'}{\exp \beta_{\max} - \exp \beta_{\max} \sum_{j=1}^{k} p_j' + \sum_{j=1}^{k} p_j'}, \tag{15}$$

which concludes the proof. □

## A.2 Numerically stable algorithm

The next proof is much simpler. Our goal is to prove

$$p_i = \exp(\log p_i' - \beta_{\max} - \log p_v' + \log p_v). \tag{16}$$

*Proof.* We begin with four facts

$$p_i = \frac{\exp \ell_i}{\sum_{j=1}^v \exp \ell_j} \quad p_i' = \frac{\exp \ell_i'}{\sum_{j=1}^v \exp \ell_j'} \quad p_v = \frac{\exp \ell_v}{\sum_{j=1}^v \exp \ell_j} \quad p_v' = \frac{\exp \ell_v}{\sum_{j=1}^v \exp \ell_j'} \tag{17}$$

which follow from the definition of softmax. Combining these, we have

$$\frac{p_i}{\exp \ell_i} = \frac{p_v}{\exp \ell_v} \quad \text{and} \quad \frac{p_i'}{\exp \ell_i'} = \frac{p_v'}{\exp \ell_v}. \tag{18}$$

Combining these again, we get

$$\frac{p_i}{p_v \exp \ell_i} = \frac{p_i'}{p_v' \exp \ell_i'}, \tag{19}$$

which we can rearrange to obtain

$$\frac{p_v' p_i}{p_i' p_v} = \frac{\exp \ell_i}{\exp \ell_i'}. \tag{20}$$

Next, we use the fact that $\exp \ell_i' = \exp \beta_{\max} \exp \ell$ to get

$$\frac{p_v' p_i}{p_i' p_v} = \exp(-\beta_{\max}). \tag{21}$$

which we can take the log of, rearrange, and exponentiate to achieve our goal

$$p_i = \exp(\log p_i' - \beta_{\max} - \log p_v' + \log p_v). \tag{22}$$

$\square$

## A.3 Stochastically robust algorithm

We would like to derive

$$\log p_v - \log p_{v-1} = \log p_v' - \log p_{v-1}'. \tag{23}$$

*Proof.* Using our result from §A.2, we have

$$p_i = \exp(\log p_i' - \beta_{\max} - \log p_v' + \log p_v) \tag{24}$$
$$p_i = \exp(\log p_i' - \beta_{\max} - \log p_{v-1}' + \log p_{v-1}). \tag{25}$$

Simply setting the right hand sides equal to one another, taking the log of both sides, then subtracting identical terms from both sides gives us our goal. $\square$

# B  Solving for full outputs using the LLM image

Given a matrix $\boldsymbol{P} \in \Delta_v^d$ whose columns span the LLM image, and $p_1, p_2, \ldots, p_d$, we can solve for $\boldsymbol{p} \in \Delta_v$. To do this, we will use the additive log ratio (alr) transform, which is an isomorphism $\Delta_v \to \mathbb{R}^{v-1}$ and is defined as

$$\text{alr}(\boldsymbol{p}) = \left(\log \frac{p_2}{p_1}, \log \frac{p_3}{p_1}, \ldots, \log \frac{p_v}{p_1}\right) \tag{26}$$

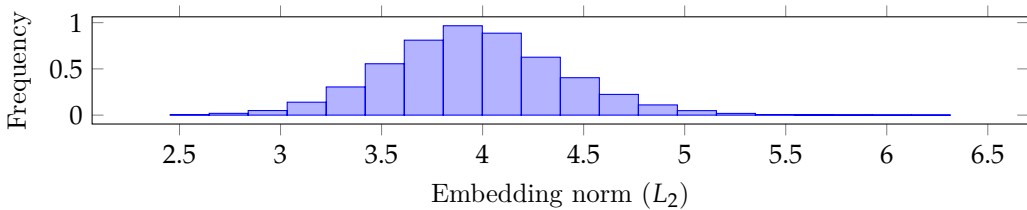

Figure 6: Softmax matrix row magnitudes (here from `pythia-70m`) are small and are distributed approximately normally within a narrow range.

to transform the columns of $P$ and $p$ into vectors in $\mathbb{R}^{v-1}$, though since we only know the first $d$ values of $p$, we can only obtain the first $d$ values of $\text{alr}(p)$. Because the alr transform is an isomorphism, we have that the columns of

$$\text{alr}(P) = \begin{bmatrix} \text{alr}(p^1) & \text{alr}(p^2) & \cdots & \text{alr}(p^d) \end{bmatrix} \in \mathbb{R}^{(v-1) \times d} \tag{27}$$

form a basis for a $d$-dimensional vector subspace of $\mathbb{R}^{v-1}$, and $\text{alr}(p)$ lies within this subspace. Therefore, there is some $x \in \mathbb{R}^d$ such that $\text{alr}(P)x = \text{alr}(p)$. To solve for $x$, all that is required is to find the unique solution to the first $d$ rows of this system of linear equations

$$\begin{bmatrix} \text{alr}(p^1)_1 & \text{alr}(p^2)_1 & \cdots & \text{alr}(p^d)_1 \\ \text{alr}(p^1)_2 & \text{alr}(p^2)_2 & \cdots & \text{alr}(p^d)_2 \\ \vdots & \vdots & \ddots & \vdots \\ \text{alr}(p^1)_d & \text{alr}(p^2)_d & \cdots & \text{alr}(p^d)_d \end{bmatrix} \begin{bmatrix} x_1 \\ x_2 \\ \vdots \\ x_d \end{bmatrix} = \begin{bmatrix} \text{alr}(p)_1 \\ \text{alr}(p)_2 \\ \vdots \\ \text{alr}(p)_d \end{bmatrix}. \tag{28}$$

After finding $x$, we can reconstruct the full LLM output $p = \text{alr}^{-1}(\text{alr}(P)x)$, where the inverse alr function is defined as

$$\text{alr}^{-1}(x) = \frac{1}{1 + \sum_{i=1}^{v-1} \exp x_i} \cdot (1, \exp x_1, \exp x_2, \ldots, \exp x_{v-1}). \tag{29}$$

