# OpenReview forum: "Logits of API-Protected LLMs Leak Proprietary Information"
_colmweb.org/COLM/2024/Conference — COLM_

### Official Review · Reviewer_gbAq · 2024-04-27

**Rating:** 7
**Confidence:** 4
**Ethics Flag:** 1

**Summary:**

This paper presents an algorithm to extract proprietary LLM information from an API-protected black-box LLM.

The algorithm uses the observation that LLM outputs are "softmax-bottlenecked", or they occupy only a low-dimensional subspace of the `R^vocab_size` output space. Using this observation, along with logit bias access to APIs, the algorithm can efficiently extract this low-dimensional subspace. This subspace can be leveraged to 1) infer the hidden dimension of the softmax layer; 2) extract a fingerprint of the API, which can be used to monitor API updates (like LLM changes).

The algorithm is tested in a real-world setting on the OpenAI GPT3.5 Turbo model, and is shown to work with high accuracy.

**Questions To Authors:**

Questions:

- What happens if the logits are postprocessed before being sent to the softmax? In particular, I think the addition of a watermark [1] could change the image of the LLM without changing the underlying LLM model. I also have a similar question for temperature changes in the API.

- How well does the method work with APIs which are not a single LLM (they route queries between two or more LLMs, like [2, 3, 4]). Does that muddy the image procured by this method, possibly adding to the estimated "effective dimension"? What if routing weights are changed? Similar question for the mixture-of-softmax method from [5].

Typos / presentation:

- The paper is missing a related work section, and I think more references would help in better grounding the work. Notably, the work is closely related to hyperparameter extraction and model extraction efforts (some examples: [6, 7, 8, 9]).

- page 3: "we turn out attention" --> "we turn our attention"

- page 3/ 4: Figure 3 is confusing and I don't think the translation to the clr space is helping my understanding (possibly better for the Appendix). Please correct me if I'm wrong, but my understanding reading the text is that everything after the softmax matmul matrix (logits, softmax, log-softmax) are in a d-dimensional subspace of R^v. Figure 3 feel like there is further reduction in image dimensionality from logits --> softmax-out (which I don't think is true).

Here's an alternative suggestion: Present a table with three columns being (transformer operation, vector space dimension, effective image dimension). Something like the table below:

(transformer operation, vector space dimension, image dimension)
(end of transformer sub-blocks, R^d, R^d),
(softmax MatMul `l = Wh`, R^V, R^d),
(softmax, `p = e^l / sum(...)`, R^V, R^d),
(log-softmax, `q = log(p)`, R^V, R^d)
etc.

Accompany this table with a simulation / points in a space where V = 3, d = 2, and one figure accompanying each of the rows above. The first row could be a 2-d plot, and all others 3-D plots with the points in a 2-d surface.

[1] - https://arxiv.org/abs/2301.10226
[2] - https://arxiv.org/pdf/2310.12963
[3] - https://arxiv.org/abs/2207.07061
[4] - https://arxiv.org/abs/2211.17192
[5] - https://arxiv.org/pdf/1711.03953
[6] - https://arxiv.org/abs/1711.01768
[7] - https://arxiv.org/abs/1802.05351
[8] - https://arxiv.org/abs/1910.12366
[9] - https://arxiv.org/abs/2305.15717

**Reasons To Accept:**

1. This paper presents technically solid work, with a very creative algorithm to extract proprietary information with high accuracy from a black-box LLM.

2. The presented attack is on a real-world LLM used by millions of people (the OpenAI GPT3.5 Turbo model), and not just in toy settings. This is quite interesting and impressive, given how secretive OpenAI has been about their model's internals. Moreover, the proposed algorithm is economically feasible (< 1000$ API credits).

3. This paper is very well-written, with a good mix of low-level linear algebra details of the algorithm, and high-level discussions on the implications of the attack.

4. The biggest implication of the method (in my opinion) is forcing more transparency from API providers on silent backend changes. Section 6 shows that even consecutive checkpoints can have very different images, which means that extracted fingerprints are fairly sensitive, and can be used to monitor API updates.

**Reasons To Reject:**

This is a great paper overall and a good fit for COLM. I have a few concerns, but overall I think the strengths outweigh the weaknesses.

1. The method (as well as prior works), rely on API providers allowing users to adjust logit biases. Based on Section 8, this method will not work without logit biases. However, I think logit bias is a fairly limited hack from OpenAI's perspective, rather than a long-term solution for controlled decoding. With rapidly improving instruction following capabilities, I think models maybe able to achieve effects similar to logit bias with just natural language instructions in the prompt, and get more natural-looking output. For reference, I don't see logit bias access to the Claude API (https://docs.anthropic.com/claude/reference/messages_post) or the Gemini API (https://ai.google.dev/api/python/google/generativeai/GenerationConfig).

2. Overall, I think the amount of information leakage from the attack is fairly limited, which limits the implications of the work. The only major information extracted is the dimensionality of the softmax matmul layer, which leaves an auditor speculating about the rest of the giant LLM (the more important proprietary information). The signature / fingerprint aspect of the attack is definitely interesting, but Section 6 and Table 2 makes me think there are too many variables that could change the API signature. This could include innocuous changes like timestamp in the hidden prompt, or search results in the hidden prompt of a RAG-enabled API. Watermarking, routed models, prompt preprocessing, output post-processors (safety filters?) could also change signatures (see questions below). Given that the APIs are rapidly changing due to industrial competition, frequent changes in API signature could leave auditors confused.

3. (minor, likely an unreasonable request) I was a bit surprised that experiments were only done on one proprietary model, and one family of open-source models. In particular, were there some cost bottlenecks to run the experiment on GPT-4 Turbo models?

---

> ### Author Rebuttal · Authors · 2024-05-30
>
> Thank you for your positive review and for noting that our findings affect millions of LLM users and could lead to better transparency and accountability for API providers.
>
> > Logit bias could go away
>
> We agree that logit bias may not be a permanent feature, and some APIs do not offer it at all. Our work remains relevant because it deals with common real-world APIs that would likely continue to have this vulnerability otherwise. Alternatively, if logit bias is useful for model auditing, it could become common practice to expose this feature as part of responsible AI deployment.
>
> > Limited information leakage, robustness of model signature for accountability
>
> Thank you for this thoughtful discussion! As this a new area of research, it is indeed an open question just how far this type of attack can go. As established in our paper, these attacks may be relatively easy to counter, but they are nonetheless important considerations when building LLM interfaces now and in the future. While we agree that our paper is mostly concerned with details about the LLM final layer, we believe this is an important first step. Indeed, we are working on a followup that gleans information by looking at deeper LLM layers.
>
> > Other models
>
> For open-source models, we found it sufficient to use Pythia models as generic stand-ins since there is no theoretical or practical reason the attack would fail with, say, GPT2. For closed-source models, we found GPT-4 turbo to be more expensive to attack, and smaller models would be of less interest. Moreover, the experiments would be considerably more expensive to run now since OpenAI has patched their API.
>
> > Post-processing outputs
>
> The watermarking method in [1] would not change these logprobs, since it affects sampling, not logprobs. Changing temperature, so long as the temperature is constant for each query, should not affect the image, since this simply scales the logit vector by a constant.
>
> > Routing models, mixture of softmax
>
> We found that, for the same query, the logprobs reported by the API were randomly selected from a set of values, leading us to develop the “stochastic logit extraction” method (Sec. 3.3). Some of the cases you mention are handled by this method. It comes down to whether each query’s output passes through the same softmax matrix. In the case of mixture-of-softmax and other nonlinear output layers, our current algorithm would not work.
>
> Thank you for your typos and suggestions. We will fix/add them.

---

> > ### Comment · Reviewer_gbAq · 2024-06-01
> > **Thank you for your comments, keeping my score**
> >
> > Thank you for your detailed replies! I will keep my original score, and continue to support the acceptance of the paper.

---

### Official Review · Reviewer_CQvU · 2024-05-06

**Rating:** 6
**Confidence:** 3
**Ethics Flag:** 1

**Summary:**

This paper discussed an interesting method to partially recover some important meta data of black box LLM, primarily the hidden dimension before embedding matrix.

The core idea is that the output logit vectors reside in dimension $d$ which is much smaller than vocabulary size $V$, thus a singular value decomposition over a collection of inferred logits matrix successfully recovers the actual hidden dimension.

The authors further discussed extensions such as inferring the model that generates certain outputs or auditing for any model updates, etc.

**Questions To Authors:**

My questions are mostly tied to "reasons to reject" section. Please respond to that section.

**Reasons To Accept:**

Though the core idea of this paper appears to be similar to [Carlini 2024], I think following are reasons to accept this paper:

1. This paper made a clearer algorithms and their complexities under different assumptions (such as w/ logprobs, numerically stable, scholastically robust, etc.)

2. This paper opened up more discussions on impact of the discovery. Including applications on detecting for any backend model change, detecting the source of the LLM mode, and ways to mitigate the stealing of meta information, etc.

3. I appreciate the authors revealed the actual number of estimated hidden dimensions of GPT3.5 (as oppose to [Carlini 2024], which is hidden from public).

**Reasons To Reject:**

The primary concern on this paper is the contribution on top of [Carlini 2024]. After comparing with two papers I feel like authors need to make it clearer on what additional value this paper brings to research community. Right now the authors did not have such comparison thoroughly. Thus, I would like the authors to include a "Contribution" section to clarify this.

Is the main breakthrough of this paper lies in the algorithmic complexity of doing LLM invocations? If so, I expect to make it clear by comparing big Os of this paper vs. [Carlini 24].

Though I appreciate the detailed discussions on potential future applications, I doubt the feasibility of quite a few. For instance "detecting the LoRA updates", the reality is people could simply skip lora updates on embedding layer; and the wording "It may be possible to detect these types of updates" sounds sneaky. It should be avoided in academia papers.

---

> ### Author Rebuttal · Authors · 2024-05-30
>
> Thank you for your helpful review and insights.
>
> > Comparison with Carlini et al. (2024) [1]
>
> We hope our work will be judged for both our **overlapping and unique contributions**, and not simply for the additional value, since we released it contemporaneously with [1]. Please note that OpenAI already patched their API when [1] came out, so we could not have conducted this research after [1]’s release. We will outline these more explicitly under a “Contributions” section.
>
> Our **overlapping** contributions with [1] include
> - Algorithm for recovering logits.
> - Theory / algorithm for recovering the output layer (the “image”).
> - Finding the embed size.
>
> Notably, [1] claims "no immediate practical consequences" of the attack, but we show several (Secs. 4, 6, 7). Our **unique** contributions include
> - Concrete estimate of OpenAI model size (5)
> - Fast logit extraction (4)
> - Model output attribution, update detection (6)
> - List of additional applications (7)
>
>
> > Primary breakthrough
>
> The primary breakthrough of our paper is our algorithm for finding the LM image (overlapping with [1]) and a collection of applications of the model image (not discussed in [1]).
>
> One of these applications is the “fast logit extraction” algorithm we present in Sec. 4, which is our unique contribution. Comparing it to our logit extraction algorithms in Sec. 3 (which are equivalent to those used in [1]) we achieve a speedup from $O(v)$ API calls to $O(d)$ API calls. Note that this new algorithm can only be used *after* the model image is found, i.e., it is not useful for finding the model image. We compare the complexity of these algorithms in Table 1. We will make these points clearer in the paper.
>
> > LoRA updates
>
> Thank you for the pointers for section 7. We agree the LoRA update idea is speculative and will revise the language to make this clearer as potential future work. Our other suggested applications (unargmaxable tokens, basis-aware sampling) are theoretically sound and well developed in their respective papers, since both algorithms take the model image as an input. The proposal in Sec. 7.3, recovering the softmax matrix, is a natural extension of finding the model image (this is also explored in [1], though their “recovered matrix” is equivalent to what we call the “model image”.)

---

> > ### Comment · Reviewer_CQvU · 2024-06-01
> > **Response**
> >
> > Thanks for your response.
> >
> > I checked the timestamps of both papers, seems indeed a concurrent case. I'll maintain my rating because 1/ COLM review guideline didn't mention how to deal with such senario. 2/ As today's APIs either do not provide logits (as Anthropic) or patched (as OpenAI), it's not very clear to me about the longer term impact. But other than that I think this paper is solid and interesting enough.

---

> > > ### Author Response · Authors · 2024-06-06
> > >
> > > As the reviewing period approaches its end we thank you for your input and reviewing our paper. Did our responses (see our above comment) address your concerns? In particular, you mentioned that your primary concern is the concurrency with Carlini et al. (2024). In light of the COLM concurrency policy, does this change your evaluation score?

---

> ### Author Response · Authors · 2024-06-01
>
> Thank you for getting back to us so quickly and giving more details about your concerns. Here is some additional relevant information.
>
> 1. Yesterday COLM organizers clarified the related work policy in an email to reviewers and authors:
> > Related work concurrency: we follow the NeurIPS policy with regards to concurrency and related work. Papers appearing less than two months before the submission deadline are generally considered concurrent to NeurIPS submissions. **Authors are not expected to compare to work that appeared only a month or two before the deadline.**
>
> 2. Both OpenAI and Cohere (not sure about others) were vulnerable to our method when we wrote the paper. OpenAI changed their API in response to becoming aware of the attack. Cohere's API was patched around the same time, presumably for the same reason. These responses indicate that these companies take our findings seriously. **In terms of longer term impact, API providers are now aware of the vulnerabilities going forward and will design their interfaces with this in mind.**
>
> Do these additional details address your concerns?

---

### Official Review · Reviewer_D4ia · 2024-05-11

**Rating:** 6
**Confidence:** 3
**Ethics Flag:** 1

**Summary:**

This paper explores the vulnerabilities in the API deployment of large language models, specifically focusing on how a limited number of API queries can expose non-public information about such models. The authors demonstrate that softmax bottleneck deduces extensive details about a model's structure, such as its embedding size. The study further suggests methods for LLM providers to shield against these vulnerabilities, while also proposing that these weaknesses could enhance transparency and accountability in LLM operations.

**Reasons To Accept:**

+ The findings are interesting, albeit it is a concurrent work to [1].

+ The techniques for

+ There are suggestions for: (1) employing this method for model auditing, and (2) mitigating this type of attack.

[1] Stealing part of a production language model, published on arXin in March 11th.

**Reasons To Reject:**

+ The attack can be easily defended by blocking logits or adding noise to the outputs.

+ Although several types of attack has be discussed in this work, but the main outcomes and purpose of this attack is unclear.
How these findings can be orchestrated as a comprehensive attack that harms LLM applications. I would consider discovering size of h in LLM is minor as it is just a number and we still need to train the models. Recovering the exact logits of the top K tokens is still minor given the 'biased' top K logits already provides essential information.

+ Improvements are needed in the clarity and organization of the writing:
  + A comprehensive overview of the algorithm and underlying theory would enhance understanding.
  + Clarification is needed on how this work differs from [1]. Specifically, if Sections 7 and 8 are intended to address these differences, a more thorough analysis in these sections could help establish a clear distinction from prior work

+ The suggested applications (in Section 7) lack sufficient detail and are not adequately supported by theoretical frameworks or experimental validation.

+ The countermeasures suggested are not substantiated by experimental evidence.

+ The paper was submitted several weeks after [1], (and arXiv version is also several days after [1]) yet it does not adequately distinguish its theoretical contributions or empirical analysis from those found in [1]. For example, an algorithm to recover the last layer was proposed in [1] but not discussed in this work. I would consider recovering the 'shape' is far less interesting than recovering 'parameters' of the linear module. This means the missing piece to [1] is the most essential part of this work.

I am aware that this is a concurrent work to [1], and my evaluation is based on the quality of this paper.

---

> ### Author Rebuttal · Authors · 2024-05-30
>
> We value your input and hope we can address your concerns:
>
> We agree that the attack is defendable, but assert that exposing it is still valuable because many LLM APIs were vulnerable to it. Otherwise, the unknown vulnerabilities would persist. As described in Sec. 8, the defenses themselves have drawbacks, which is likely why OpenAI’s recent patch kept logit bias.
>
> > Outcomes/purpose of attack. Harmful?
>
> Our attack reveals the LM image, which has several applications listed in the Secs. 4-7. We agree (last par., Sec. 9) that the "attack" is not egregiously harmful to LLM providers (though OpenAI likely hides model details for a reason). Providers may opt to *keep* these "vulnerabilities" to build trust, given the useful, benign capabilities we propose (e.g., detecting model updates.)
>
> > Distinction from [1]
>
> Our **overlapping** contributions with [1] include
> - Algorithm for recovering logits.
> - Theory / algorithm for recovering the output layer (the “image”).
> - Finding the embed size.
>
> Notably, [1] claims "no immediate practical consequences" of the attack, but we show several (Secs. 4, 6, 7). Our **unique** contributions include
> - Concrete estimate of OpenAI model size (5)
> - Fast logit extraction (4)
> - Model output attribution, update detection (6)
> - List of additional applications (7)
>
> We will outline these more explicitly in a “Contributions” section. We hope our work will be judged for both our **overlapping and unique contributions**, since we released it contemporaneously with [1]. Please note that OpenAI already patched their API when [1] came out, so we could not have conducted this research after [1]’s release.
>
> > Recovering last layer
>
> We would like to clarify that our “model image” corresponds to the “last layer” recovered in [1], both of which are linear transformations of the parameters in the last layer. We will make this explicit.
>
> > Suggested application (Sec. 7).
>
> We use Sec. 7 as “future work” to show that there are many applications (beyond the paper’s scope) aside from those detailed in Secs. 4-6. We will edit to convey this clearly.
>
> > Countermeasure experiments
>
> We cite (first par., Sec. 8) Morris et al. (2023) who already demonstrate logprob-free logit extraction. Other countermeasures block our algorithm, so it is unclear what experiment to run.
>
> > Theory, overview of algo.
>
> Thank you, we will more clearly highlight our algorithms in our edits. Do you have specific feedback on how we could make the theory more comprehensive?

---

> ### Comment · Reviewer_D4ia · 2024-06-04
> **Thanks for your response**
>
> We have to admit that the concurrent work [1] is more preferred and mature, because it 1) involves more solid theories, 2) is motivated by a concrete attack goal and corresponding threat model, and 3) is published several days earlier than this submission. I consider this work concurrent to [1] and my decision is independent to 3.
>
> I still do not feel the properties proposed in this work pose a solid threat to LLM, but they are interesting properties to NLP community (probably worth discussing in COLM). I would suggest improving this work to form a solid attack, but considering the timing, I am not against the work to be published to COLM for the awareness of such threats in NLP community.

---

### Decision · Program_Chairs · 2024-07-10

**Decision:**

Accept

**Comment:**

The paper studies how the logits of a black-box LLM (such as OpenAI’s models) can reveal non-public information about the model. The key idea is that most modern LLMs’ outputs are limited to a low-dimensional subspace of the full output space. Combined with this paper’s another contribution on an improved algorithm to recover the full probability distribution with limited API output, it provides a very efficient way to recover the softmax layer and the rank of it, which consequently leads to various non-public information about the model such as the hidden dimension of the model (which could hint the model size) and the nature of the model updates (e.g., whether or not the softmax layer was updated).

Reviewers acknowledged that the paper is technically solid and made a clear algorithmic contribution under various assumptions such as the availability of logprobs, stochastic changes, etc  (Reviewer CQvU, gbAq). Reviewers also think the paper opens up a broader discussion on forcing more transparency from API providers on silent backend changes, employing this method for model auditing, and developing mitigation for this type of attacks (Reviewer CQvU, gbAq). Reviewers also think the findings are interesting and the revealed information about OpenAI models is valuable (Reviewer CQvU, gbAq, D4ia).

Some concerns have been raised regarding the practical implications of the findings, given that the attacks are defensible, the information it leaks are limited (Reviewer gbAq, D4ia), and it relies on specific assumptions like the presence of logit bias features (Reviewer gbAq).

[At least one review was discounted during the decision process due to quality]